# Inaccurate Risk Assessment by the ACS NSQIP Risk Calculator in Aortic Surgery

**DOI:** 10.3390/jcm10225426

**Published:** 2021-11-20

**Authors:** Tessa M. Hers, Jan Van Schaik, Niels Keekstra, Hein Putter, Jaap F. Hamming, Joost R. Van Der Vorst

**Affiliations:** 1Department of Surgery, Leiden University Medical Centre (LUMC), 2333 ZA Leiden, The Netherlands; t.m.hers@lumc.nl (T.M.H.); j.van_schaik@lumc.nl (J.V.S.); n.keekstra@lumc.nl (N.K.); j.f.hamming@lumc.nl (J.F.H.); 2Department of Medical Statistics and Bioinformatics, Leiden University Medical Centre (LUMC), 2333 ZA Leiden, The Netherlands; h.putter@lumc.nl

**Keywords:** risk assessment, risk prediction model, ACS NSQIP risk calculator, vascular surgery

## Abstract

Objectives: The aim of this retrospective study was to assess the predictive performance of the American College of Surgeons (ACS) risk calculator for aortic aneurysm repair for the patient population of a Dutch tertiary referral hospital. Methods: This retrospective study included all patients who underwent elective endovascular or open aortic aneurysm repair at our institution between the years 2013 and 2019. Preoperative patient demographics and postoperative complication data were collected, and individual risk assessments were generated using five different current procedural terminology (CPT) codes. Receiver operating characteristic (ROC) curves, calibration plots, Brier scores, and Index of Prediction Accuracy (IPA) values were generated to evaluate the predictive performance of the ACS risk calculator in terms of discrimination and calibration. Results: Two hundred thirty-four patients who underwent elective endovascular or open aortic aneurysm repair were identified. Only five out of thirteen risk predictions were found to be sufficiently discriminative. Furthermore, the ACS risk calculator showed a structurally insufficient calibration. Most Brier scores were close to 0; however, comparison to a null model though IPA-scores showed the predictions generated by the ACS risk calculator to be inaccurate. Overall, the ACS risk calculator showed a consistent underestimation of the risk of complications. Conclusions: The ACS risk calculator proved to be inaccurate within the framework of endovascular and open aortic aneurysm repair in our medical center. To minimize the effects of patient selection and cultural differences, multicenter collaboration is necessary to assess the performance of the ACS risk calculator in aortic surgery.

## 1. Introduction

As stated in the declaration on the promotion of patient’s rights of the World Health Organization (WHO) of 1994, patients have the right to be fully informed about the potential risks and benefits of each procedure [1]. Consequently, risk assessment is one of the cornerstones of informed consent and shared decision making. It is, therefore, of paramount importance, in particular in preoperative consultations [2,3,4]. 

Preoperative risk assessment can also contribute to risk reduction by improving preoperative consultation and work-up. Moreover, it can improve postoperative management since it permits better preparation and planning among treatment teams, it is valuable in patient expectation management and can even provide risk-adjusted comparison of surgical outcomes [4,5,6,7,8,9,10,11]. To this day there is no consensus on what constitutes sufficient preoperative risk assessment. The estimation of risks of postoperative complications that is shared with the patient varies between treating surgeons, depending mainly on experience [4,5]. Therefore, there is an increasing focus on risk stratification tools [9].

A statistically reliable predictive model improving the quality of preoperative risk assessment is potentially of great value. The American College of Surgeons National Surgical Quality Improvement Program (ACS NSQIP) developed a risk prediction model which takes twenty-one patient factors into consideration when predicting the risks of postoperative adverse events for various procedures. The risk predictive model developed by the ACS is based on a great number of cases of which 9.7% concern vascular procedures and represents an important step forward in this domain. It is simple to use and easily accessible online [5].

There are several studies that examine the reliability of the ACS risk calculator within plastic- and reconstructive surgery, surgical oncology, neurosurgery and acute surgery [6,12,13,14,15,16,17,18]. These predominantly single center studies show inconsistent results, because the ACS risk calculator is predictive for most complications, but there is under- or overestimation of the actual risks. Within aortic surgery the ACS risk calculator has not been evaluated yet.

Aneurysmal degeneration is, after atherosclerosis, the most common disease of the aorta. The natural history of aortic aneurysms differs between a subclinical course and lethal rupture, with a worldwide mortality of 0.2 million [19,20,21]. Currently 74% of aortic aneurysms patients are treated by infrarenal endovascular aneurysm repair (EVAR). The remaining treatment options of aortic aneurysms consist of complex endovascular procedures (thoracic, fenestrated or branched endografting) or open surgical repair (OSR) [21,22,23,24]. 

The aim of this retrospective study is to assess the validity of the ACS risk calculator for aortic aneurysm repair for the patient population in a tertiary referral university medical center.

## 2. Materials and Methods

### 2.1. Study Design

The study was approved by the Institutional Review Board as a retrospective review of all patients who underwent elective aortic surgery in our medical center from January 2013 to December 2019.

Data extracted from the electronic patient files included demographics (i.e., age, sex, BMI, functional status, smoking status), medical comorbidities, and postoperative complications within thirty days of the procedure, as used in the ACS risk calculator. Patient demographics were manually entered into the risk calculator to generate an individual risk assessment. The Current Procedural Terminology (CPT) codes used were, 34,803 = EVAR, 0087T = (Fenestrated/Branched) EVAR, 33,881 = (Thoracic) EVAR, 35,081 = OSR using tube prosthesis, and 35,102 = OSR using bifurcated prosthesis. The option to view Geriatric Outcomes was not used and “Surgeon adjustment of Risk” was not altered to minimize bias. For each case the estimated risk percentages for ‘serious complication’, ‘any complication’, pneumonia, cardiac complication, surgical site infection, urinary tract infection, venous thromboembolism, renal failure, readmission, return to operating room, discharge to nursing or rehab facility, sepsis, predicted length of hospital stay were obtained.

### 2.2. Statistical Analysis

Demographic data were summarized. Statistical analyses to assess the performance of the ACS risk calculator were completed for the overall study population. Subset analysis was not performed due to the relatively low number of cases per procedure. The ACS risk calculator does not calculate an estimated risk of renal failure for patients who are preoperatively suffering from acute renal failure or patients who are preoperatively in need of dialysis. Therefore, patients with these specific comorbidities were excluded from the statistical analysis for the predicted risk of renal failure (n = 13). Additionally, some of the included complex endovascular procedures (thoracic/fenestrated/branched EVAR) were intentionally performed in two sessions rather than one consecutive operation to reduce the risk of spinal ischemia. For data capture of these procedures the decision was made to only record the complications and length of hospital stay of the first, often more impactful, session (n = 6). The area under the receiver operating characteristic (ROC) curves, calibration plots, Brier scores, and Index of Prediction Accuracy (IPA) values were generated with R version 4.0.3. 

### 2.3. Discrimination—ROC Curve

To assess the discriminative performance of the ACS risk calculator, i.e., verifying whether patients with a complication actually had a higher risk prediction than those without that specific complication, the ROC curve (or concordance statistic) was used. The ROC curve, plotting sensitivity (true positive rate) against 1—specificity (false positive rate), results in an area under the curve (AUC) between 0.5 (equal to chance) and 1.0 (perfect discrimination) [25]. AUC values > 0.7 are generally considered sufficient [12,14,16,18].

### 2.4. Calibration 

Calibration assesses to what extent the predicted risk corresponds to the observed proportion [25,26]. Calibration plots were used to visualize calibration with the predicted risks on the x-axis and the observed frequencies on the y-axis. Perfect predictions lie on the diagonal curve. If the predictions lie below the diagonal curve, this indicates that the prediction model overestimates risk. If predictions lie above the diagonal curve, the prediction model underestimates risk [27].

### 2.5. Brier Score and Index of Prediction Accuracy (IPA) Values

To quantify how close predictions were to the actual outcome, the Brier score was used. This takes both discrimination and calibration into account. A Brier score of 0 is representative of a perfect model. For comparison purposes, a null model Brier score was calculated, which contains no predictions and thereby assesses the overall prevalence of outcome in the dataset [25]. 

IPA values were generated in order to compare the Brier score to the null model Brier score. The IPA values were determined by the formula 1—(model Brier score/null model Brier score), where an IPA value of 1 represents a perfect model, 0 a non-informative model and < 0 harmful models (higher prediction error than null model) [28].

### 2.6. Length of Hospital Stay

Due to the numerical nature of length of hospital stay, a scatter plot with regression line was made to analyze the accuracy of the ACS predicted length of hospital stay. Ideally the regression line would be y = x.

## 3. Results

Three hundred and fifty-three patients underwent aortic surgery between 2013 and 2019 in our medical center. Two hundred thirty-four patients were identified congruent with one of the included CPT codes. Table 1 shows patient demographics. The majority of the patients were male (85.9%), >65 years old (81.1%) and were classified as ASA class II or III (96.2%).

The overall 30-days postoperative complication rate was 40.6% with a mortality rate of 3.1%. The most frequent complications were renal failure (*n* = 21, 9.5%) and discharge to nursing or rehab facility (*n* = 22, 9.4%). An overview of statistical outcomes is provided in Table 2.

### 3.1. Discrimination

The area under the ROC curve (AUC) shows that five out of thirteen risk predictions were sufficiently discriminative. These include urinary tract infection (AUC = 0.757), venous thromboembolism (AUC = 0.797), readmission (AUC = 0.737), discharge to nursing or rehab facility (AUC = 0.806), and sepsis (AUC = 0.700). The majority of the AUC, however, are < 0.7.

### 3.2. Calibration

The calculator structurally underestimates the risk of complications in this dataset. Except for the risk of readmission (6.8% predicted vs. 6.0% observed) and the risk of sepsis (2.6% vs. 1.7%), which were overestimated. This is reflected in the calibration plots in Figure 1.

### 3.3. Brier Score and IPA Values

Although most of the Brier scores are close to 0, the IPA values indicate that the risk calculator is far from perfect and might even lead to a harmful model, with seven out of thirteen IPA values < 0. These include the IPA values for serious complication (IPA = −0.084), any complication (IPA = −0.249), pneumonia (IPA = −0.039), surgical site infection (IPA = −0.027), urinary tract infection (IPA = −0.051), return to operating room (IPA = −0.035) and sepsis (IPA = −0.035). The highest IPA value found is for the risk of death (IPA = 0.118).

### 3.4. Length of Hospital Stay

The scatter plot and regression line found for the length of hospital stay is shown in Figure 2. The regression line equals y = 4.04 + 1.55 * x. The risk calculator structurally underestimates the length of hospital stay in our study population.

## 4. Discussion

The ACS risk calculator underestimates the risk of aortic surgery in the present investigated patient group. Vaziri et al. [12] mentions that the ACS risk estimates are focused on single CPT codes. As stated above, some complex aortic aneurysm procedures are split into two sessions, giving patients the time to recover between surgeries. It is unclear whether the ACS risk calculator takes the possibility of two hospital admissions around one CPT code into account. Therefore, a single CPT code is perhaps not sufficient to calculate adequate risk prediction for complex procedures; this might be one of the explanations for the findings of the current study. 

Likewise, Vosler et al. [16] propounded the theory that the complexity of procedures included in ACS validation studies, which are substantially affected by surgical skill, case dependent variables and level of postoperative care, is a possible reason for inaccuracy of the ACS risk calculator. Presumably the ACS risk calculator is ideally used within the framework of low-complex procedures. This study took place in a tertiary referral medical center with patients referred for high complex aneurysm repair which could have an effect on the risk of complications. The surgeons and intervention radiologists involved in the procedures all had significant experience with technically complex aneurysm repair.

Another explanation for the conflicting results in contemporary literature and the results found in this study is the lack of procedure-specific metrics. Some studies show that adding specialty or procedure specific surgical risk estimations improve the predictive accuracy of the ACS risk calculator [6,29]. Although the ACS risk calculator can be seen as an extensive calculator, it will never capture the full set of relevant patient characteristics. For example, no data on the use of anticoagulants, pathophysiology of the aneurysm, aneurysmal diameter or clamp positioning and level of distal anastomosis for OSR are implemented in the calculation. Addition of operation specific variables would improve prediction and enable procedure specific complication estimations. Unfortunately, this would probably lead to various calculators with different algorithms which defeats the intended purpose of the ACS risk calculator as general and widely applicable [8]. 

This study has several strengths, including the number of cases with an event, i.e., ninety-five. As Collins et al. [30] recommends an ideal sample size for predictive model validation with around a hundred cases with an event. Furthermore, it is the first study to validate the ACS risk calculator for aortic aneurysm repair. This study is also distinctive in that it evaluates discrimination and calibration separately. Most studies only analyze the discriminative abilities and Brier score of the ACS risk calculator. Calibration itself is at least as important as discrimination, potentially being the source of the divergence observed [8,26,31]. Moreover, this study is one of the two studies that compares the ACS Brier score to a null model Brier score [15]. Finally, our study is the first to calculate an IPA value for an accurate comparison of Brier scores.

Although this study confirms findings from previous publications, its conclusions need to be weighed with caution given the single center nature of the study. According to van Calster et al. [26] calibration is affected by heterogeneity between treating hospitals. Liu et al. [7] and Cohen et al. [8] emphasize that discrimination improves by heterogeneity of patient population and procedure type. 

Despite the benefits of the ACS risk calculator, it may not be applicable for our patient population undergoing high complex aortic aneurysm repair. Further research is necessary to assess the predictive performance of the ACS risk calculator for other aortic surgery groups, preferably for separate CPT codes. Additionally, multicenter development of a procedure specific risk calculator might improve predictive accuracy and permit the estimation of procedure specific complications.

## 5. Conclusions

This study demonstrates the inaccuracy of the ACS risk calculator within the framework of endovascular and open aortic aneurysm repair in patients treated in a tertiary referral medical center. Further research is needed in a multicenter setting to further evaluate the predictive performance of the ACS risk calculator. 

## Figures and Tables

**Figure 1 jcm-10-05426-f001:**
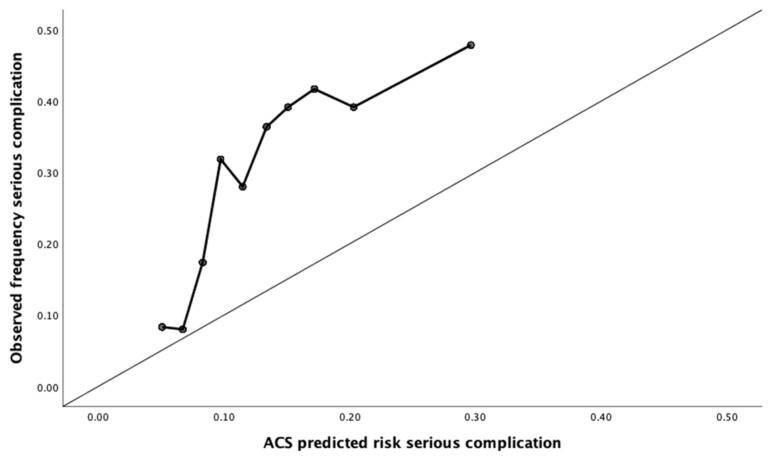
Calibration plots for the risk of complications.

**Figure 2 jcm-10-05426-f002:**
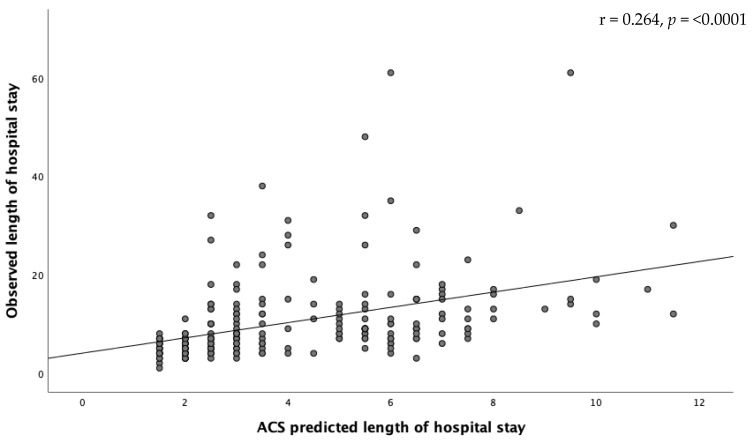
Scatter plot and regression line for the length of hospital stay.

**Table 1 jcm-10-05426-t001:** Demographic.

	Value (*n* = 234)
Age, *n* (%)	
Under 65 years	44 (18.8)
65–74 years	104 (44.4)
75–84 years	77 (32.9)
85 years or older	9 (3.8)
Gender: Male, n (%)	201 (85.9)
Procedure, n (%)	
Open tube	35 (15.0)
Open bifurcation	44 (18.8)
EVAR ^1^	77 (32.9)
(F/B)EVAR ^2^	56 (23.9)
TEVAR ^3^	22 (9.4)
Height, mean (SD), m	1.76 (0.09)
Weight, mean (SD), kg	81.22 (14.73)
BMI, mean (SD)	26.09 (4.10)
Functional status, *n* (%)	
Independent	229 (97.9)
Partially dependent	5 (2.1)
Dependent	0 (0)
ASA, *n* (%)	
I	1 (0.4)
II	112 (47.9)
III	113 (48.3)
IV	8 (3.4)
Steroid use for chronic condition: Yes, *n* (%)	12 (5.1)
Ascites within 30 days prior to surgery: Yes, *n* (%)	1 (0.4)
Systemic sepsis within 48 hours prior to surgery: Yes, *n* (%)	0 (0)
Ventilator dependent: Yes, *n* (%)	0 (0)
Disseminated Cancer: Yes, *n* (%)	5 (2.1)
Diabetes, *n* (%)	
No	195 (83.3)
Oral Medication	32 (13.7)
Insulin	7 (3.0)
Hypertension requiring medication: Yes, *n* (%)	163 (69.7)
Congestive heart failure in 30 days prior to surgery: Yes, *n* (%)	9 (3.8)
Dyspnea, *n* (%)	
None	194 (82.9)
With moderate exertion	39 (16.7)
At rest	1 (0.4)
Current smoker (within 1 year): Yes, *n* (%)	87 (37.2)
History of severe COPD: Yes, *n* (%)	31 (13.2)
Dialysis: Yes, *n* (%)	4 (1.7)
Acute renal failure: Yes, *n* (%)	9 (3.8)

^1^ EVAR, endovascular aortic aneurysm repair; ^2^ (F/B)EVAR, Fenestrated/Branched EVAR; ^3^ TEVAR, Thoracic EVAR; BMI, Body Mass Index; COPD, Chronic Obstructive Pulmonary Disease.

**Table 2 jcm-10-05426-t002:** Comparison of observed outcomes to ACS calculated risks.

Complications	Observed Outcomes, %	Predicted Risks, Mean %	AUC	Brier Score	Nullmodel Brier Score	IPA Values
Serious complication	29.5	13.6	0.680	0.225	0.208	−0.084
Any complication	40.6	14.1	0.661	0.301	0.241	−0.249
Pneumonia	8.5	2.5	0.621	0.081	0.078	−0.039
Cardiac complication	2.1	2.0	0.562	0.021	0.021	0.000
Surgical site infection	3.8	1.6	0.560	0.038	0.037	−0.027
Urinary tract infection	8.5	1.3	0.757	0.082	0.078	−0.051
Venoust hromboembolism	1.3	0.9	0.797	0.013	0.013	0.000
Renal failure *	9.5	7.4	0.657	0.089	0.086	0.018
Readmission	6.0	6.8	0.737	0.055	0.056	0.000
Return to operation room	6.8	5.4	0.535	0.064	0.064	−0.035
Death	3.1	1.3	0.554	0.030	0.029	0.118
Discharge to nursing or rehab facility	9.4	7.1	0.806	0.075	0.085	0.000
Sepsis	1.7	2.6	0.700	0.017	0.017	−0.035

AUC, area under the curve; IPA, Index of Prediction Accuracy. * *n* = 221.

## Data Availability

The data presented in this study are available on request from the corresponding author. The data are not publicly available due to legal restrictions.

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
