# Peer review of "Inaccurate Risk Assessment by the ACS NSQIP Risk Calculator in Aortic Surgery"

_jcm, 2021, doi:10.3390/jcm10225426_

Round 1
Reviewer 1 Report
The authors present a well written report that evaluates the performance of the ACS risk calculator in patients treated for abdominal aortic aneurysms. The methodology is sound and fits the research questions of the study.
My main concern is about the selected study population. The authors chose to include all patients planned for elective aortic surgery, including those with infrarenal, thoraco-abdominal or thoracic aortic aneursms, etc. This makes the studied population rather heterogenous and several important factors that could have a major impact on outcomes and complication risks are not incorporated in the risk prediction model. The table with demographic data does not inform the reader of these included subgroups. This is a major limitation of the findings of the study and should at least me addressed more in the discussion section. As the authors suggest in the discussion section multicenter evaluation per subgroup of aortic procedures/aneurysm levels would be an interesting follow-up study.
Reviewer 2 Report
This manuscript examiners the effectiveness of a standard risk calculator provided by the American College of Surgeons, for prognostication, both to inform patient selection for operation and to inform patients of likely risk of the procedure. The authors have specifically examined the risk calculator for aortic surgery. The results support the conclusion that the ACS risk calculator significantly underestimates observed risk in their cohort. These data are of substantial interest and relevance to aortic surgeons in their clinical practice. Furthermore, the specific issues raised concerning the effectiveness of individual parameters in predicting specific risk would be of benefit for any future review of the risk calculator by ACS.
I had a few specific minor comments and corrections for the authors to consider:
- Line 85 - spelling - prosthesis
- Line 93 - data were summarized.
- A scatter plot of length of hospital stay with a fitted regression line is provided - could the authors provide an r value and a p value for the fit of the regression line?
- Figure 1 calibration plots - the font size for the axis labels is very small and difficult to read
- Line 237: The authors could make a general comment about their particular hospital setting (beyond their mention of a tertiary referral hospital), in the context of their comments of the relevance of surgical skill. For example, were all vascular surgeons involved in the study appropriately specialized in the procedures undertaken and did they have appropriate currency of practice?
- Line 251: risk calculator as a general and widely applicable.
- Line 252: ...several strengths, including the....
